# Bioactive Components of Areca Nut: An Overview of Their Positive Impacts Targeting Different Organs

**DOI:** 10.3390/nu16050695

**Published:** 2024-02-29

**Authors:** Huihui Sun, Wenzhen Yu, Hu Li, Xiaosong Hu, Xiaofei Wang

**Affiliations:** 1College of Food Science and Nutritional Engineering, China Agricultural University, Beijing 100083, China; huihui.sun@cau.edu.cn (H.S.); yuwz9903@163.com (W.Y.); huxiaos@263.net (X.H.); 2Sanya Institute of China Agricultural University, Sanya 572025, China; 3Department of Entomology, College of Plant Protection, China Agricultural University, Beijing 100083, China; lih@cau.edu.cn

**Keywords:** areca nut, bioactive components, pharmacological functions

## Abstract

*Areca catechu* L. is a widely cultivated tropical crop in Southeast Asia, and its fruit, areca nut, has been consumed as a traditional Chinese medicinal material for more than 10,000 years, although it has recently attracted widespread attention due to potential hazards. Areca nut holds a significant position in traditional medicine in many areas and ranks first among the four southern medicines in China. Numerous bioactive compounds have been identified in areca nuts, including alkaloids, polyphenols, polysaccharides, and fatty acids, which exhibit diverse bioactive functions, such as anti-bacterial, deworming, anti-viral, anti-oxidant, anti-inflammatory, and anti-tumor effects. Furthermore, they also display beneficial impacts targeting the nervous, digestive, and endocrine systems. This review summarizes the pharmacological functions and underlying mechanisms of the bioactive ingredients in areca nut. This helps to ascertain the beneficial components of areca nut, discover its medicinal potential, and guide the utilization of the areca nut.

## 1. Introduction

*Areca catechu* L. is a tropical crop belonging to the Arecaceae, which includes 181 genera and 2600 species [1]. It is widely distributed, and its fruit areca nut has a long history of being consumed in South and Southeast Asia, including China, Indonesia, Malaysia, Myanmar, Bangladesh, and India [2]. There are more than 600 million areca nut chewers worldwide [3], and the number is still growing. The huge economic benefits have made *A. catechu* an important cash crop and source of livelihood for millions of farmers in its planting area. By 2016, the scale of *A. catechu* planting had reached 0.38 million acres in Hainan Province, China, and the production value was over USD 40 billion [4].

Generally, Areca fruit consists of two main parts, namely, the husk and kernel [5]. They are used separately or together for edible or medicinal purposes, while there are several ways to eat areca nuts. In general, areca nut can be chewed alone or be wrapped inside the Piper betel leaf with or without slaked lime, tobacco, spices, sweeteners, or some other constituents, and can be processed into commercialized areca nut products as well [6,7]. The processed areca husks usually go through several processing steps like soaking, enzymatic hydrolysis, drying, and adding bitterness, during which their average hardness reaches 10^5^/g [8], and the hard woody texture makes them liable to cause mechanical damage during chewing [9]. Areca nut and betel quid have been classified as group 1 carcinogens by the International Agency for Cancer Research (IARC) for the risk of oral and esophageal cancers [7], causing a great impact on the areca nut industry.

As a matter of fact, areca nut is one of the four major southern medicines in China [10] and is recorded in the Compendium of Materia Medica [11]. Since 1953, areca nut has been included in the Pharmacopoeia of China [12]. Many prescriptions are recorded for the medicinal value of different forms of areca nut, including the decoction pieces of areca nut, Charred Semen Arecae, and arecae pericarpium [13]. Their therapeutic effects on beriberi edema, tenesmus, malaria, abdominal pain, indigestion, and diarrhea have been frequently documented [14]. In addition, areca nut can also treat periodontitis, premature ejaculation, difficulty urinating, and glaucoma [15]. Additionally, areca nut combined with other medicinal materials also has important effects, such as the famous compound medicine Da Yuan Yin, which is used to treat multiple inflammation-related diseases [16], of which areca nut can dissipate phlegm, resolve masses, and relieve dampness and damp evil [17]. These suggest that there are many bioactive components in areca nuts worth exploiting. Modern studies have isolated and identified the bioactive components of areca nut, including arecoline, procyanidin, luteolin, and catechin. They possess a variety of pharmacological activities, such as anti-inflammatory, anti-depression, anti-tumor, hypoglycemic, and anti-oxidant effects [18,19,20].

In this review, we summarized the research progress on the bioactive constituents and pharmacological activities of areca nut and put forward comprehensive insights into various beneficial functions of bioactive components in areca nut, with the aim being to provide theoretical guidance to the upgrading of the areca nut industry and providing a theoretical basis for its application in medicines and functional foods.

## 2. Main Chemical Components of Areca Nut

Areca nut has a complex chemical composition, and multiple components in different parts of it are detected (Figure 1). The major constituents of areca nut are polyphenols (10–30%), polysaccharides (18–25%), fibers (10–15%), fatty acids (10–15%), and alkaloids (0.3–0.7%) [21,22]. Areca nut husk is primarily composed of cellulose and contains more lignin, which makes it have higher hardness and produce more mechanical friction on the oral mucosa during chewing, but it can be an excellent wood fiber resource material [8,23]. Apart from that, there are a variety of bioactive substances in areca nut (Figure 1).

### 2.1. Alkaloids

Alkaloids are nitrogenous organic compounds commonly found in higher plants, such as Papaveraceae, Loganiaceae, Leguminosae, Ranunculaceae, and Menispermaceae [24]. Most of them are cyclic nitrogenous compounds and natural psychoactive substances [25,26]. The total alkaloid content of areca nut is 0.3–0.7% [27]. The four major alkaloids identified in areca nut are arecoline (*N*-methyl-1,2,5,6-tetrahydropyridine-3-carboxylic acid methyl ester), arecaidine, guvacoline, and guvacine. Their content ranges in fresh seeds are 0.30–0.63%, 0.31–0.66%, 0.03–0.06%, and 0.19–0.72%, respectively [28]. Arecoline is an acid-based amphoteric compound belonging to pyridine alkaloids [29], which was first isolated from areca nut in 1888 by German pharmacist E. Jahns [22]. It is often combined with tannins, so it is usually extracted by immersion in ammonia first and then extracted using ethanol, ether, or chloroform [30]. Moreover, it presents a similar structure to nicotine, which has strong stimulative effects. After alcohol, caffeine, and nicotine, arecoline has already become the fourth most widely used addictive substance around the world [31].

Interestingly, although arecoline is always the richest alkaloid in the young green areca nut extracts [32], the concentration of guvacine is almost three times higher than arecoline in mature areca nuts [33], which suggests that the contents of different alkaloids in areca nuts vary with ripeness. In addition, five new alkaloids, named arecatemines A-C [34] and acatechu A and B [35], were isolated from the nuts and the dried fruit of *A. catechu*, respectively. However, some unknown alkaloids in areca nut may still exist, and their potential physiological activities need to be studied in future research.

### 2.2. Polyphenols

The phenolic compounds or polyphenols are secondary metabolites of plants and the most prevalent anti-oxidant phytochemicals [36]. Areca nut is rich in phenolics, including flavonoids and tannins [27,37]. Total phenols and flavonoids are significantly higher in areca nut than in areca husk and areca flower [38]. Besides this, the total amount of phenolic substances in areca nut varies with ripeness, length, and place of origin. For example, areca nuts with higher ripeness or longer lengths were found to be richer in phenolics [39].

#### 2.2.1. Flavonoids

Flavonoids refer to compounds that consist of two phenolic hydroxyl benzene rings connected by three central carbon atoms [40,41], which are widely found in the roots, stems, leaves, flowers, and fruits of various plants [42]. They are also identified and purified from many Chinese medicinal materials, such as pueraria, ginkgo biloba, and sea buckthorn [43,44,45]. Plenty of studies have shown that flavonoids are capable of inhibiting atherosclerosis and are regarded as potential anti-cancer, anti-viral, anti-inflammatory, and anti-oxidant components [46,47,48,49,50].

Areca nut is also rich in flavonoids, mainly including catechins, isorhamnetin, quercetin, liquiritigenin, 5,7,4′-trihydroxy-3′,5′-dimethoxydihydroflavone, and chrysoeriol [51]. Ma et al. used 80% ethanol to extract the catechins in areca nut under heating (80 °C) and reflux three times, and then the eluent was collected and recovered under reduced pressure. Finally, the highest content of catechins was obtained (5.9%) after vacuum-drying at 60 °C [52]. Recently, two flavonoids, calquiquelignan M and calquiquelignan N, were isolated from the areca nut [53]. Besides this, luteolin, 4′,5-dihydroxy-3′,5′,7-trimethoxyflavonone, jacareubin, flavan-3-ol, flavan-3,4-diols, and glycyrrhizin have also been identified [54].

#### 2.2.2. Tannins

Tannins are a class of water-soluble polyphenolic compounds with complex structures [55], which can be found in legume seeds, cereals, cacao, vegetables, coffee, tea, berries, and nuts [56,57]. They are flavanol derivatives and are mainly categorized into hydrolyzable tannins and condensed tannins [58]. The tannins in areca nuts are mainly condensed tannins, which are more abundant in unripe areca nuts and decrease with ripening [59,60]. Tannins usually taste astringent and bitter, so they can make areca nuts taste bitter as well [61]. Tannins are usually extracted by solvent extraction, which can be done using water, ethanol, methanol, chloroform, acetone, etc. Studies have optimized extraction conditions for the tannins in areca nut, using 80% acetone, adjusting the pH to 4, and extracting for 90 min [62]. The tannins extracted from areca nut in previous studies include (+)-catechin, (−)-epicatechin, procyanidin B1, procyanidin B2, areca tannin A1, areca tannin B1, areca tannin B2, and areca tannin C1 [51,63]. The procyanidins in areca nut can be extracted using an acetone–aqueous solution by ultrasonication at room temperature with the ultrasonic power of 600 W, and the final extraction rate reached 3.41% according to Chen et al. [64]. Various biological functions of tannins have been discovered, such as anti-oxidant, anti-bacterial, anti-viral, anti-parasitic, anti-inflammatory, and anti-diarrheal activities [65]. Correspondingly, these functions may explain the pharmacological activities of areca nut.

### 2.3. Polysaccharides, Lipids, and Other Components

Areca nut also contains polysaccharides, triterpenes, steroids, fatty acids, and other components. The polysaccharides are abundant in areca nut [66]. A novel neutral polysaccharide, PAP1b, which was isolated from areca nut and mainly composed of mannose, galactose, arabinose, and xylose, possesses excellent scavenging ability on DPPH and hydroxyl radicals, suggesting its anti-oxidant potential [11]. Meanwhile, such traits can be used for inflammation treatment, anti-aging, and the prevention of cardiovascular disease.

The triterpenes and steroids in areca nut include arborinol, fernenol, arundoin, cycloartenol, and arborinol methyl ether [27,67]. The fatty acids mainly found in areca nut are lauric acid, myristic acid, palmitic acid, stearic acid, and oleic acid [27,51]. In addition, some trace elements have also been identified in areca nut, such as Zn, Fe, Al, Cr, Co, Mn, Cu, and so on [68].

Furthermore, there are still ingredients in areca nut that remain to be discovered. In addition to the above substances, volatile components and metabolites remain to be explored.

## 3. Functional Effects of Areca Nut Components

Plant-derived bioactive components have been widely investigated, and some have identified extensive pharmacological functions and biological activities. Bioactive components in areca nuts have also been shown to have a variety of pharmacologically beneficial effects, mainly including effects on the nervous, digestive, and endocrine systems, as well as anti-inflammatory, anti-tumor, anti-oxidant, anti-bacterial, deworming, and anti-viral effects (Figure 2 and Table 1).

### 3.1. Effects on the Nervous System

#### 3.1.1. Refreshing Functions

Chewing areca nut can lead to flushing, palpitations, and sweating. Meanwhile, it can relax the body, enhance well-being, maintain concentration, improve alertness, and increase work efficiency [81,107]. For this reason, long-distance drivers often chew areca nuts to stay awake and relieve work stress [108]. These effects may be attributed to the alkaloids present in areca nut, particularly arecoline. Arecoline stimulates the parasympathetic nerves and promotes excitation [31]. It acts as a muscarinic receptor agonist and easily crosses the blood–brain barrier to activate central M receptors, thereby stimulating the HPA axis [83]. Arecoline has been found to significantly irritate VTA dopaminergic neurons, which can produce dopamine to achieve an excitatory effect [85] and is associated with neurodegenerative diseases [109,110,111], indicating that the arecoline in areca nut may have the potential to be developed for the treatment of neurodegenerative diseases.

Moreover, arecaidine and guvacine are inhibitors of gamma-aminobutyric acid (GABA) uptake and can bring a sensation of euphoria [107]. Phenolic substances also contribute to stimulation, for they can stimulate the release of catecholamines, including norepinephrine, epinephrine, and dopamine [112]. This class of neurotransmitters can promote intense excitability [113].

#### 3.1.2. Anti-Depression Effects

Areca nut has been used to treat depression for years in Mongolia, and there is a traditional Mongolian herbal formula containing areca nut called Areca Thirteen Pill, also named Gao You-13 (GY-13), which has an excellent anti-depression effect. GY-13 improves depression by modulating the chemokine/chemokine receptor axis [114]. Besides, this GY-13 can significantly restore the expression level of cyclic adenosine monophosphate (cAMP), protein kinase A (PKA), brain-derived neurotrophic factor (BDNF), and cAMP response element binding (CREB), and increase the proliferative activity in the hippocampus, thereby improving depressive behaviors [115]. Yao et al. proved that areca nut can improve depressive behavior by activating the BDNF signaling pathway as well [116].

Depression is also closely related to the deficiency of monoamine neurotransmitters (e.g., norepinephrine and serotonin) [117] or the excessive activity of monoamine oxidases (MAOs), which have two subtypes, MAO-A and MAO-B [118]. Their inhibition contributes to reducing the breakdown of dopamine, tryptamine, and tyramine so as to achieve anti-depressant functions [119]. Areca nut extracts can boost the secretion of monoamine neurotransmitters, including dopamine, serotonin, and norepinephrine, achieving an anti-depressant effect [69]. In addition, acute arecoline can increase serotonin and norepinephrine levels in zebrafish brains [81]. This property is very similar to that of hypericum perforatum [120], which has been used for the treatment of depression for more than 100 years, indicating that areca nut may be promising for developing anti-depressant drugs [121]. Besides this, aqueous ethanolic extract, hexane fraction, and aqueous fractions of areca nut can inhibit MAOs. Among them, aqueous fractions have the best inhibitory effects, with an IC_50_ value of 20 µg/mL [70]. Furthermore, isorhamnetin, chrysoeriol, luteolin, and chrysophanol in the husk and seeds of *A. catechu* have been identified as MAO-A inhibitors, so they also have potential anti-depressant efficacy [104]. Meanwhile, MAO inhibitors can be used for the treatment of Parkinson disease [122,123], suggesting that areca nut extracts may also have the potential for use in anti-Parkinson drugs development.

#### 3.1.3. Analgesic Efficacy

Areca nut shows significant anti-nociceptive effects as well. The total alkaloids show dose-dependent effects in suppressing formalin-induced pain at doses of 100–400 mg/kg, which contributes to the inhibition of cyclooxygenase-2 (COX-2) expression [80]. Furthermore, Areca nut extracts and one of the ingredients, Procyanidins, can relieve migraines, which have a complex pathogenesis, mainly including dura mater inflammation and plasma protein leakage from the blood vessels [124,125]. Via pretreatment with areca nut extracts orally at doses of 250 and 500 mg/kg, plasma protein extravasation is reduced by 16.21% and 23.24%, respectively [59]. The intrathecal administration of areca nut can reduce spinal nerve ligation and chemotherapy-induced neuropathic pain, which works through the α-2 adrenoceptors and 5-HT7 receptors, on account of their contribution to the antiallodynic effects of areca nut [72].

Both pain and depression are associated with monoamine transmitters, so depression has been linked to analgesia as well [126]. It is precisely because areca nut extracts can increase the levels of neurotransmitters and BDNF [69,115] that they can achieve the analgesic effect as well, indicating that areca nut may exert a dual function in abirritation and anti-depressant. As a matter of fact, studies have indicated that 5-HT receptors and COX are targets in Alzheimer’s disease (AD) as well, so areca nut extracts may also have an effect on AD [127,128].

#### 3.1.4. Treatment of Alzheimer’s Disease

The prevalence of Alzheimer’s disease (AD) is increasing worldwide, which has significant implications for individuals and society. Multiple therapeutic targets have been discovered, including muscarinic receptor, glycogen synthase kinase 3 beta, *N*-methyl-D-aspartate receptor, histamine receptor, MAO, and acetylcholinesterase [127]. Areca nut shows promise in the treatment of AD, which is characterized by the decline of cognition and memory loss. The acetylcholine muscarinic receptors, especially M1 and M4, are generally recognized as the hinge of neuromental control and cognition [129]. Brown et al. successfully developed HTL9936, an M1-receptor orthostatic partial agonist, which improved cognition and memory in mice and beagles [130]. Similarly, as a muscarinic receptor agonist, arecoline is easier to combine with M1, M2, M3, and M4 receptors. This may explain its ability to improve memory retention in Alzheimer’s patients [83]. Moreover, areca nut extract exhibits anti-acetylcholinesterase activity, thereby increasing acetylcholine levels and contributing to memory formation and cognitive function [131,132]. Besides this, MAO-induced changes in brain monoamine neurotransmitter levels are also associated with AD. Thus, acting as an MAO inhibitor, areca nut extract may also potentially decrease the expression of MAO-B in platelets and the brain in AD patients [133].

Furthermore, another important obstacle to AD therapy is that over 95% of neurotherapeutic drugs cannot pass through the blood–brain barrier, which significantly limits their efficacy [134]. Fortunately, arecoline has been proven to pass through the blood–brain barrier. Notably, FDA-approved AD drugs include memantine, galantamine, donepezil, and rivastigmine, and they are all single-target drugs [135]. Compared to them, areca nut is expected to be developed into an efficient multi-target drug for AD.

#### 3.1.5. Relieve Schizophrenia

Areca nut and its isolated compounds also have a therapeutic effect on schizophrenia. A survey into schizophrenia or schizoaffective disorder found that chewing areca nuts (more than 7.5 areca nuts per day) can relieve the symptoms compared with those who chew areca nuts in small amounts, or non-chewers [136]. Arecoline has also been demonstrated to alleviate schizophrenia symptoms by reducing demyelinating and spatial working memory impairment in mice [87]. In addition, dysfunctional GABA interneuron activity is considered to be the core mechanism underlying cognitive dysfunction in schizophrenia [137]. As inhibitors of GABA, arecaidine and guvacine may serve as modulators in schizophrenia as well.

#### 3.1.6. Relieve Epilepsy

In addition, chewing areca nuts can reduce seizure frequency in people with epilepsy. Of the 152 patients, areca nut chewers showed a 58.7% reduction in the frequency of seizures. Compared with those non-chewers, the number of epileptic seizures in areca nut chewers decreased by an average of 2.1 per month [106]. Meanwhile, guvacine, as a potent inhibitor of GABA uptake, has been demonstrated to be useful for the treatment of seizure disorders [138], and there are studies on the addition of lipophilic groups to the ring nitrogen of guvacine to make anticonvulsant drugs [139].

### 3.2. Effects on the Digestive System

Chewing areca nut can boost the secretion of saliva, stimulate the gallbladder muscles, and accelerate the expulsion of bile, therefore contributing to food digestion [140,141]. Areca nut can also stimulate M receptors and increase gastrointestinal smooth muscle tone and intestinal peristalsis. It can act on cholinergic M3 receptors, increasing the average amplitude of the circular smooth muscle contraction wave [77]. Moreover, Yao et al. showed that the moderate addition of arecoline to the diet could up-regulate M3 mRNA levels and increase digestive enzyme activities to aid digestion and absorption [92].

Charred Semen Arecae is a traditional Chinese medicine for bloating and constipation treatment [142]. Both Semen Arecae and Charred Semen Arecae could increase gastric function and improve gastric motility and emptying. Their mechanisms of action are dependent on the reduction in cholecystokinin mRNA expressions in the intestine and hypothalamus, which could increase serum levels of substance P and motilin [78]. Further comparisons found that Charred Semen Arecae had a more significant effect on gastrointestinal motility than areca nut, possibly due to the Maillard reaction products. [143]. Besides this, arecoline can promote the amplitude of smooth muscle contraction of the isolated small intestine in rabbits [144]. Therefore, it is speculated that arecoline may be the active ingredient that promotes digestion in Semen Arecae.

Moreover, areca nut is also effective against gastroenteritis. The pharmacopeia records that areca nut can be used to treat malaria, diarrhea, and tenesmus. The oral administration of areca-derived polyphenols can reduce intestinal inflammation. To be specific, 0.05% doses of areca-derived polyphenols can reduce the infiltration and degranulation of duodenal mast cells and diarrhea in mice, thereby reducing allergic reactions after ovalbumin-induction [100]. Aside from oral administration, acupoint application can also work. The acupoint application therapy of modified Wuzhuyu Binglang Tang can relieve chronic non-atrophic gastritis symptoms by reducing the release of inflammatory cytokines and regulating gastrointestinal hormones, which could protect gastric mucosa and enhance gastrointestinal function [145]. Consequently, areca nut can be used to develop drugs for promoting digestion as well as improving gastrointestinal inflammation.

### 3.3. Effects on the Endocrine System

#### 3.3.1. Hypoglycemic Effects

Extracts from various parts of areca nut exhibit hypoglycemic activity. Accordingly, applying 10 µg/mL procyanidins in areca nut extract can inhibit cyclic adenosine monophosphate (cAMP)/dexamethasone-induced gluconeogenesis by 40%, related to the inhibition of the glucose-6-phosphatase (G6Pase) and phosphoenolpyruvate carboxykinase (PEPCK) expression, which is equivalent to the effect of 10 nM insulin [101]. Furthermore, polyphenols or polysaccharides in the areca nut extract may relieve the symptoms of diabetes by inhibiting α-glucosidase activity [95]. In addition, type 2 diabetes is often complicated by hypertension, and the ratio is as high as 73.6% of adult patients, which increases the risk of cardiovascular disease and microalbuminuria [146]. The polyphenols in areca nut have been demonstrated to lower blood pressure, possibly lowering blood sugar while avoiding complications [147]. Previous work has indicated that polyphenols can improve complications of diabetes such as diabetic nephropathy, cardiovascular disease, neuropathy, and diabetic retinopathy [148], and natural polyphenols are safer even at high doses than synthetic and semi-synthetic drugs [149]. Therefore, we perceived that polyphenols in areca nut extract might be a good choice for diabetes treatment as well. The current drugs used to treat diabetes have certain side effects, such as metformin causing nausea and diarrhea in 25% of patients [150]. The pharmacopeia states that areca nut can treat diarrhea simultaneously, so better diabetes drugs may be developed from areca nut, though it is imperative to isolate and purify the key active ingredients, which is essential for clinical applications.

#### 3.3.2. Hypolipidemic Effects

Many studies have shown that cholesterol levels are strongly associated with cardiovascular disease, cancer, and Alzheimer’s disease [151]. Nevertheless, existing lipid-lowering drugs have some drawbacks. For example, PCSK9 inhibitors and statins work to make blood cholesterol enter the liver, which may increase the burden on the liver [152]. Areca nut extract has been shown to decrease the total serum cholesterol, serum triglycerides, and the atherosclerotic index [71]. It is achieved by inhibiting intestinal acyl-CoA:cholesterol acyltransferase (ACAT) and pancreatic cholesterol esterase (pCEase) activity, which can reduce cholesterol in plasma by 25% and increase the excretion of cholesterol in feces, thus avoiding burdening the liver [73]. Therefore, we believe that areca nut extract is promising for use in lipid-lowering. Nevertheless, research on the hypolipidemic effects of areca nut is still insufficient. It is necessary to use more animal models to verify the lipid-lowering effect of areca nut and elucidate its mechanism of regulating blood lipids further.

#### 3.3.3. Effects on Hormone Levels

Areca nut has been proven effective in regulating the levels of some endocrine hormones. Arecoline can stimulate thyroid activity, mediated by muscarinic cholinergic receptors in the short term, while the long-term consumption of arecoline will reduce the concentrations of thyroxine (T_4_) and triiodothyronine (T_3_) in serum and increase serum thyroid-stimulating hormone (TSH) in mice [89]. T_3_ has been shown to have neuroprotective and anti-inflammatory effects and inhibit the progression of pulmonary fibrosis in silicosis [153,154,155]. Experimental evidence has showed that a large amount of desquamated cell debris could be seen in the thyroid–follicular lumen of mice after long-term arecoline consumption. Therefore, paying more attention to the appropriate intake dose and treatment duration of arecoline is indispensable for its clinical application.

### 3.4. Anti-Inflammatory Effects

Inflammation is an autoimmune disorder and the pathological basis of many diseases [156]. Experimental evidence shows that ethyl acetate, hexane, and aqueous fraction in areca nut extracts can suppress carrageenan-induced edema. Among them, the aqueous fraction works best, which can inhibit over 80% edema at a dose of 100 mg/kg after 2 h of treatment, much better than aspirin [19], which is a kind of non-steroidal anti-inflammatory drug often used in clinical treatment [157,158,159].

Areca nut extracts exhibit anti-inflammatory effects through multiple pathways. Therein, the aqueous fraction especially exerts anti-inflammatory effects due to the inhibition of arachidonic acid metabolism and the degradation and/or inactivation of inflammatory mediator prostaglandin E2 (PGE2) [19]. Areca nut polysaccharide can inhibit the lipopolysaccharide-induced NO production to alleviate inflammation [105]. Moreover, ferulic acid, catechin, and quercetin in areca nut have been detected to have an excellent scavenging ability of reactive oxygen species (ROS) [160], whose overexpression can cause an inflammatory response [161]. Thus, areca nut extracts may have a potential medical benefit as an anti-inflammation agent.

However, most current research remains focused on mixed extracts, and the specific active ingredients in areca nut are still unclear. The further identification and purification of the active ingredients in areca nut and in-depth studies of their anti-inflammatory mechanisms will provide an important basis for anti-inflammatory drug development and clinical application.

### 3.5. Anti-Osteoporotic Effects

Chewing areca nuts can reduce the risk of osteoporosis [162]. Therefore, polyphenols isolated from areca nut have anti-osteoporotic properties in ovariectomized rats, which can increase bone mass by downregulating 5-HT and promoting bone resorption and formation [96]. Sun et al. showed that areca nut polyphenols can induce the proliferation, differentiation, and mineralization of osteoblasts in vitro [37]. Furthermore, Liu et al. found that arecoline can promote osteoblast differentiation and inhibit osteoclastogenesis [93]. Considering that osteoporosis and cardiovascular disease are common side effects caused by long-term aspirin treatment for inflammation [163], areca nut extracts are set to be developed into better drugs to treat inflammation and avoid or reduce side effects.

### 3.6. Anti-Tumor Efficacy

Tumors have the characteristics of infinite proliferation and apoptosis disorders, and are the most threatening diseases around the world. Clinical apoptosis induction is a vital tumor treatment strategy [164]. Areca nut extract can be used to fight against hepatocellular carcinoma, basal cell carcinoma, and some other cancers, mainly by inducing apoptosis. Among them, areca nut extract can effectively inhibit the proliferation of HepG2, HepJ5, and Mahlavu cells, with an IC_50_ value of 30–48 µg/mL [76]. This is probably because areca nut extract can induce the production of ROS, which could trigger mitochondrial damage and arrest the cell cycle [165]. Meanwhile, areca nut extract treatment induces the activation of autophagy and increases lysosomal formation, thereby achieving anti-hepatocellular carcinoma effects [76]. It is worth noting that flavonoids have been found to have similar anti-tumor mechanisms, including arresting the cell cycle, inducing autophagy and apoptosis, and regulating ROS-scavenging enzyme activity [166]. So, it is speculated that the flavonoid levels in areca nut, which are much higher than in some previously reported plant extracts, may contribute to its anti-cancer effects.

The arecoline in areca nut can reduce the secretion of IL-6 in tumor cells, increase the level of p53, and induce apoptosis. Therefore, the effects of areca nut on the growth of tumor cells may be related to the expression level of cytokines. For example, due to various expression levels of IL-6 in different cells, arecoline has no effect on the human keratinocytes line HaCaT, but can suppress the growth of BCC-1/KMC tumors [90]. Besides this, p53 is a powerful tumor suppressor that regulates cell division and coordinates the expression of target genes, such as p21, p48, Bax, and VKORC1L1, which can promote DNA repair, cell cycle arrest, and apoptosis [167,168,169,170,171,172,173]. P53 mutations occur in more than 50% of tumors, and studies have shown that reactivating fractional p53 is sufficient to repel cancer, so p53 is an attractive target for cancer [174,175,176]. Yan et al. also demonstrated that arecoline can alter genes involved in the p53 signaling pathway [177]. Therefore, it is speculated that areca nut may have the potential to inhibit a variety of tumors by increasing the expression of p53.

In the future, in-depth studies of the optimal dose and formulation are expected to guide the development of the anti-cancer drug. Importantly, the high production of areca nut has the potential to reduce the cost of anti-cancer drugs.

### 3.7. Anti-Oxidant Effects

Areca nut polyphenols show significant anti-oxidant activity in lipopolysaccharides (LPS)-stimulated RAW264.7 cells, mainly by inhibiting the MAPK pathway and activating the Nrf2/HO-1 pathway to reduce ROS generation [98]. Major polyphenols isolated from the areca nut, including epicatechin and syringic acid, can exhibit excellent anti-oxidant activity as well [36]. Experimental evidence shows that polyphenols extracted from *A. catechu* husk have a good anti-fatigue function [97], and oral administration of areca nut procyanidins can prevent UVB-induced photoaging [102]. Besides this, the polysaccharide from areca nut PAP1b shows anti-oxidant activity as well, especially in scavenging DPPH and hydroxyl radicals [11]. Moreover, areca nut has been found to be a potential collagenase inhibitor, which has been experimentally proven to have an anti-wrinkle effect related to its anti-oxidant properties [178,179]. In addition, anti-oxidants have demonstrated therapeutic potential in cancer for their redox homeostasis capacity, suggesting the promising anti-cancer effect of areca nut extracts. However, an inappropriate dose, duration, and cancer type could also cause side effects [39,180,181].

### 3.8. Anti-Bacterial Efficacy

In areca nut extract, fernenol, arundoin, and the mixture of stigmasterol and β-sitosterol can inhibit the growth of *Colletotrichum gloeosporioides*. So they can be used to prevent and control the postharvest anthracnose disease in mangoes, even better than the commercial fungicide benomyl [182]. The methanol extract of areca nut has an anti-bacterial effect on Gram-positive and Gram-negative bacteria, and is especially effective against the Gram-negative bacteria *Escherichia coli* [18]. The volatile components of areca nut have anti-bacterial properties as well. Machová et al. extracted the components from areca nut with simultaneous hydrodistillation extraction (SHDE), finding that the SHDE extract had the strongest bacteriostatic effect on *Streptococcus canis*, from which 98 volatile compounds were further isolated and identified, including aliphatic hydrocarbons, alcohols, fatty acids, carbonyl compounds, esters, terpenoids, terpenes, and arecoline [86]. In addition, areca nut extract can have a synergistic anti-bacterial effect with other substances. For example, when forming the silver nanoparticles (AgNPs) combined with silver nitrate, areca nut extract exhibited better inhibitory activity against antibiotic-sensitive and drug-resistant bacteria [183]. Besides this, mixed extracts of areca nut and *Punica granatum* L. can inhibit the biofilm formation and metabolism of *Staphylococcus aureus*, *Escherichia coli*, *Enterobacter aerogenes*, and *Salmonella enteric*. Thereby, the ethanolic extract showed the strongest inhibitory impacts [184].

### 3.9. Deworming Efficacy

Areca nut has a relatively long history of use as a traditional medicine to treat parasitic diseases such as taeniasis. For example, areca nut extract can reduce the infection rate of chicken coccidiosis and its mucosal damage by inhibiting the causative agent *Eimeria tenella* [75,185]. Moreover, it can increase the concentration of interleukin-2 (IL2), which can be used for the prevention and treatment of autoimmune diseases and can improve intestinal inflammation [75,186,187]. In addition, research has demonstrated that the condensed tannins in multiple plant extracts have deworming effects [188]. Thus, condensed tannins extracted from areca nut may also be effective repellents. Besides this, areca nut ethylene extract can protect livestock from being infringed by helminth parasitism by suppressing the activity of fumarate reductase and succinate dehydrogenase, which affects the carbohydrate metabolism of parasites [74]. Furthermore, areca nut extract can inhibit gastrointestinal parasitic nematodes [79], which have widespread impacts on livestock health, indicating that it can be effective in animal husbandry [189].

### 3.10. Anti-Viral Efficacy

Phenolics in areca nut may have anti-viral effects. For instance, procyanidin B1, arecatannin A1, arecatannin B1, and arecatannin B1 in areca nut have anti-HIV activity via inhibiting the HIV type 1 protease [99]. Besides this, quercetin in areca nut may have the ability to inhibit COVID-19 replication [190].

## 4. The Potential Health Risks of Areca Nut

It has to be noted that areca nut is considered to represent a risk due to its association with oral and esophageal cancers [7]. Firstly, areca nut husk is rich in crude fiber, which makes it hard and rough in texture, thus chewing areca nut would produce long-term mechanical irritation on the oral mucosa, which has been proposed as a pivotal risk factor for oral cancer [23,191,192]. Secondly, areca nut extracts and arecoline are found to have cytotoxicity and carcinogenic effects on human oral mucosal fibroblasts and gingival fibroblasts in vitro [193]. Specifically, 50 μg/mL arecoline exposure shows up as cytotoxic to human gingival fibroblasts [194]. Besides this, the median lethal dose (LD_50_) of arecoline in mice was 174 mg/kg (i.g.) [28]. It is worth noting that low-dose arecoline (6 mg/kg, i.g.) reduced serum and hepatic lipid levels in mice, while high-dose arecoline (30 mg/kg, i.g.) promoted oxidative stress in the liver [195], which demonstrates that arecoline intake confers both pharmacological and toxicological properties, depending on the dosage of intake. Importantly, areca nut chewers are always addicted to it. This is primarily due to arecoline, which presents a similar structure to nicotine and has already become the fourth most widely used addictive substance around the world [31]. Therefore, although arecoline showed promising potential in treating neurodegenerative diseases, the long-term high-dose intake of it may lead to toxic effects, such as oral submucous fibrosis, cytotoxicity, and genotoxicity [28]. Taking into account the potential harm and addiction, the application of arecoline must be strictly regulated. In addition, previous research has reported that several supplements, such as *N*-acetylcysteine or antioxidants like vitamins, manifest cancer risk reduction. For instance, *N*-acetylcysteine can conjugate with arecoline and produce mercapturic acid compounds, which are relatively less toxic and excreted through urine [196]. Thus, the combined use may be a promising strategy to reduce the potential toxicities and side effects of long-term arecoline use.

## 5. Conclusions and Future Prospects

Areca nut contains various bioactive components, such as alkaloids, polyphenols, polysaccharides, and fatty acids. These bioactive ingredients possess multiple health benefits, including anti-depression, anti-inflammatory, anti-tumor, gastrointestinal protection, neuroprotection, hypoglycemic, hypolipidemic, anti-oxidant, anti-bacterial, deworming, and anti-viral effects, indicating that areca nut has high medicinal value and its processing has broad application prospects.

However, most current research on areca nut focuses on complex extracts, and research on pure substances is often limited to alkaloids or polyphenols. Thus, a large number of specific bioactive ingredients are still worthy of further study. Meanwhile, isolating and purifying these bioactive constituents is essential to improving their utilization, and will be conducive to studying their underlying mechanisms. Besides this, using more reliable animal models and high-level validation experiments to conduct a risk assessment is also very important. Finally, it is essential to choose the proper dosage and delivery method of clinical treatment, and so the recommended dosages and detailed mechanisms underlying the efficacy of these bioactive ingredients in areca nut deserve to be deeply explored, which will be conducive to developing related drugs and upgrading the areca nut industry.

## Figures and Tables

**Figure 1 nutrients-16-00695-f001:**
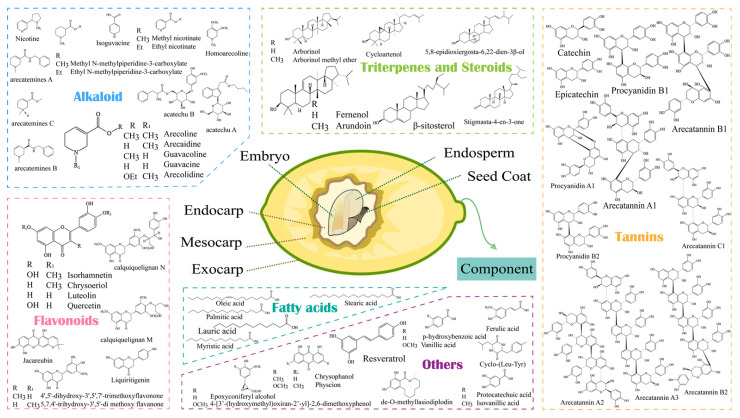
Structure and chemical composition of areca nut. Areca nut consists of pericarp (exocarp, mesocarp, and endocarp) and seed (seed coat, endosperm, and embryo). The chemical components identified in areca nut mainly include alkaloids, flavonoids, tannins, fatty acids, triterpenes, and steroids.

**Figure 2 nutrients-16-00695-f002:**
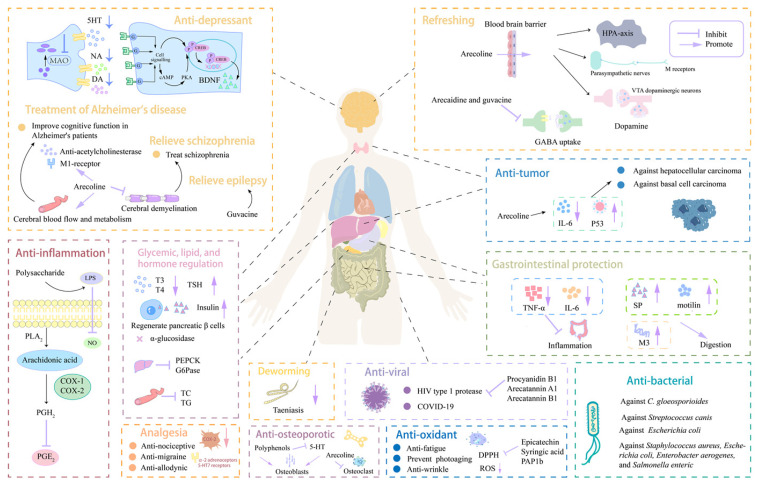
Summary of the beneficial functions of areca nut components. Areca nut exhibits diverse pharmacological activities, including effects on the nervous system (refreshing, anti-depression, analgesia, treatment of Alzheimer’s disease, relieving schizophrenia and epilepsy), effects on the endocrine system (glycemic, lipid, and hormone regulation), and gastrointestinal protection, anti-inflammatory, anti-tumor, anti-oxidant, anti-bacterial, deworming, and anti-viral effects. The sharp arrow indicates activation, and the flat arrow indicates inhibition.

**Table 1 nutrients-16-00695-t001:** Summary of the beneficial functions and mechanisms of areca nut extracts and isolated bioactive compounds.

Extracts/Ingredients		Functions	Model	Mechanism	Dose	Reference
Areca nut extract	Aqueous fraction	Anti-inflammatory	Mice and rats	↓ Arachidonic acid metabolism, ↑ degradation and/or inactivation of PGE2	100 mg/kg (p.o.)	[19]
	Aqueous fraction	Anti-depressant	Rats	↑ Secretion of serotonin	10 mg/kg (i.p.)	[69]
	Hexane fraction	Anti-depressant	Mice and rats	↓ MAOs	2.5 mg/kg (i.p.)	[70]
	70% aqueous methanol extract	Hypolipidemic	Rats	↓ pCEase and pancrelipase	10 mg/kg (i.g.)	[71]
	80% (*v*/*v*) aqueous ethanol extract	Anti-allodynic	Rats	↑ α-2 adrenoceptors and 5-HT7 receptors	300 µg (intrathecal administration)	[72]
	90% aqueous methanol extract	Hypolipidemic	Rats	↓ ACAT and pCEase activity	0.5% (p.o.)	[73]
	Methanol extract	Anti-bacterial	In vitro	↑ Destroying the cell wall		[18]
	Ethylene extract	Deworming	In vitro	↓ Carbohydrate metabolism of the parasite		[74]
	Areca nut extract	Deworming	Chicks	↓ The causative agent *Eimeria tenella*	100 mg/kg (p.o.)	[75]
		Anti-tumor	Mice	↑ ROS, autophagy, and lysosomal formation	20 mg/kg (i.p.)	[76]
	Pericarpium arecae extract	The improvement of intestinal peristalsis	Guinea pigs	↑ Cholinergic M3 receptors	1 g/mL concentrated stock solution of crude drug (3% concentration)	[77]
	Semen Arecae and Charred Semen Arecae extract	Gastrointestinal protection	Rts	↓ Cholecystokinin mRNA expressions, ↑ serum levels of substance P and motilin	3 g/kg (p.o.)	[78]
	Areca nuts powder extracted with supercritical carbon dioxide	Deworming	Chicks	↓ Eimeria tenella	100 mg/kg (p.o.)	[75]
	Ethylene extract	Deworming	In vitro	↓ Carbohydrate metabolism	0.5 mg/mL	[74]
	Crude aqueous-methanol extract	Deworming	Sheep	↓ Hatching of parasite eggs, the development of the infected larval stage, and the physiological functions of the parasite	0.33 g/kg (p.o.)	[79]
Alkaloids	Total alkaloids	Analgesia	Mice	↓ COX-2 expression	100 mg/kg (i.g.)	[80]
	Arecoline	Anxiolytic	Zebrafish	↑ Norepinephrine, ↑ Serotonin	10 mg/mL	[81]
		Refreshing		↑ Parasympathetic nerves		[82]
			Zebrafish	↑ Central M receptors	0.01 ppm	[83]
			Rats	↑ HPA axis	0.2 mg/kg (i.p.)	[84]
			Rats	↑ VTA dopaminergic neurons	0.2 mg/kg (i.v.)	[85]
		Anti-bacterial	In vitro	↓ Bacillus subtilis, Enterococcus faecalis, Escherichia coli, Pseudomonas aeruginosa, Staphylococcus aureus, Streptococcus		[86]
		Relieve schizophrenia	Mice	↓ Cerebral demyelination	2.5 mg/kg/day (p.o.)	[87]
		Relieve Alzheimer’s disease	Rats	↑ Cerebral blood flow and metabolism	2 mg/kg (p.o.)	[88]
		Hormone levels regulation	Mice	↑ Muscarinic cholinergic receptors	10 mg/kg (i.p.)	[89]
		Anti-tumor	BCC-1/KMC and HaCaT	↓ IL-6, ↑ P53 and apoptosis	30 μg/mL	[90]
			In vitro	Alters the cell cycle, ↓ cell viability, ↓ cancerous prostate cells	0.4 mM	[91]
		Gastrointestinal protection	Grass carp	↑ M3, ↑ Keap1a/Nrf2 signaling pathway, ↓ RhoA/ROCK signaling pathway	1 mg/kg (p.o.)	[92]
		Anti-osteoporotic	Mice	↑ Osteoblast differentiation, ↓ osteoclastogenesis	5 mg/kg (p.o.)	[93]
	Arecaidine	Refreshing	In vitro	↓ GABA uptake		[94]
	Guvacine					
Phenols	Total phenols	Hypoglycemic	In vitro	↓ α-glucosidase activity	IC_50_ 1.50 ± 0.31 μg/mL	[95]
		Anti-osteoporotic	Rats	↓ 5-HT, ↑ bone resorption and formation	400 mg/kg (i.g.)	[96]
			In vitro	↑ Proliferation, differentiation, and mineralization of osteoblasts	25 µg/mL	[37]
		Anti-fatigue	Mice	↑ Lactate dehydrogenase, superoxide dismutase, and catalase activities, ↓ malondialdehyde content	20 mg/kg (i.g.)	[97]
		Anti-oxidant	RAW264.7 cells	↓ MAPK pathway, ↑ Nrf2/HO-1 anti-oxidant pathways	40 µg/mL	[98]
		Anti-viral	In vitro	↓ HIV type 1 protease	0.2 mg/mL	[99]
	Procyanidins	Analgesia	Mice and rats	↓ Plasma protein extravasation and inflammation in the dura	250 mg/kg (p.o.)	[58]
		Gastrointestinal protection	Mice	↓ Th2 responses, ↑ induction function of myeloid-derived suppressor cells	0.05% in water	[100]
		Hypoglycemic	Mice	↓ G6Pase and PEPCK	10 mg/kg/day (i.g.)	[101]
		Anti-photoaging	Mice	↓ Cyclooxygenase-2, matrix metalloproteinase, and oxidative stress	10 mg/kg (p.o.)	[102]
		Anti-hypertensive	In vitro	↓ Angiotensin-converting enzyme	IC_50_ 1.51 ± 0.65 mg/mL	[103]
		Anti-viral	In vitro	↓ HIV type 1 protease	0.2 mg/mL	[99]
	Isorhamnetin	Anti-depressant	Mice	↓ MAOs	0.2 mL/10 g (i.g., concentration of 2.5 mg/mL)	[104]
	Chrysoeriol Luteolin Chrysophanol					
	Syringic acid	Anti-oxidant	In vitro	↓ DPPH radical scavenging activity, hydroxyl radical scavenging activity, and reducing power	EC_50_ 0.409 mg/mL and 0.188 mg/mL, respectively	[36]
	Epicatechin					
		Anti-hypertensive	In vitro	↓ Angiotensin-converting enzyme	IC_50_ 1.51 ± 0.65 mg/mL	[103]
	Catechin					
	Arecatannin	Anti-viral	In vitro	↓ HIV type 1 protease	0.2 mg/mL	[99]
Triterpenes and Steroids	Fernenol	Hypoglycemic	In vitro	↓ α-glucosidase activity	IC_50_ 1.50 ± 0.31 μg/mL	[95]
	Arundoin				EC_50_ 47.5 mg/L	
	The mixture of stigmasterol and β-sitosterol				EC_50_ 56.7 mg/L	
Polysaccharide		Anti-inflammation	Raw264.7 cell	↓ NO production	IC_50_ 85.64 mg/mL	[105]
		Anti-oxidant	In vitro	↓ DPPH/hydroxyl radicals	2 mg/mL	[11]
Fatty acids		Anti-bacterial	In vitro	↓ Bacillus subtilis, Enterococcus faecalis, Escherichia coli, Pseudomonas aeruginosa, Staphylococcus aureus, Streptococcus		[86]
Areca fruit		Relieve epilepsy	Human	↑ Muscarinic receptor	Three nuts/day	[106]

↑: Activated. ↓: Inhibited. p.o.: Oral administration. i.g.: Oral gavage. i.p.: Intraperitoneal injection. i.v.: Intravenous injection. EC_50_: Half maximum effective concentration. IC_50_: Half maximal inhibitory concentration.

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
