# Peer review of "Bioactive Components of Areca Nut: An Overview of Their Positive Impacts Targeting Different Organs"

_nutrients, 2024, doi:10.3390/nu16050695_

Round 1
Reviewer 1 Report
Comments and Suggestions for Authors
Xiaofei et al. submitted the manuscript entitled: Bioactive components of areca nut: An overview of their positive impacts targeting different organs, in which they summarized natural products as well as their biofunctions identified in areca nut. It’s a very informative review and the potential readers may be interested in this topic.
My suggestions are as follows:
1. Section 3.1, it seemed that the authors used large amount of secondary references (e.g. line 180, statement is irrelevant with ref. 70), which significantly decrease the reliability of these statements. The authors are suggested to identify the conclusions and results in original references and replace all the secondary references in section 3: functional effects of areca nut components.
2. Directly describe the bioactivity of natural components and avoid using statement like: Areca has a bioactivity of XXX, if applicable. For example, line 262: the statement is Areca nut also has a therapeutic effect on XXX, but the ref. 106 is actually talking about one component, Quetiapine.
3. Table 1: for GY-13 and Wuzhuyu Binglang Tang, are there any evidences that the bioactivities are (mostly) from areca? If yes, please cite them. If no, the authors are suggested to remove these two compounds.
4. Conclusion and future prospects: the authors are encouraged to move line 93-95 into this section and discuss about the potential harm of areca.
5. Some typos (but not limited to them): line 219, thereinto; line 151-152, please check if calcium is one of trace elements.
Comments on the Quality of English Language
Some typos are mentioned in the previous comments. The authors are suggested to check throughout the whole manuscript.
Reviewer 2 Report
Comments and Suggestions for Authors
The manuscript is well-written and has interesting data from areca nut biological activities. The review aimed to assess the beneficial effects of the bioactive compounds of the areca nut on different organs. However, I have concerns about the isolated bioactive compounds relationship described and the areca nut. Furthermore, the authors very succinctly mentioned the carcinogenic potential of areca nut.
Many studies about medicinal plants have demonstrated that the beneficial effects caused by them are a result of all bioactive compounds' synergic action. Therefore, the authors were very general in relating the effects of the isolated bioactive compounds to the areca nut actions. We observed this generalism by looking at the references used to describe some of the isolated bioactive compound effects, which have nothing to do with areca nut, such as 68, 71, 72, and 190, among others, reinforcing that these effects cannot be directly related to areca nut. Therefore, the authors should reformulate the review highlighting the effects of areca nut extracts demonstrated at the beginning of Table 1, showing the differences between the different extracts and the essential oil, the bioactive compounds, and the beneficial effects found. Furthermore, the authors should remove from Table 1 the effects of the extract that mixes areca nut with pumpkin seeds, as the actions found may be related to pumpkin seeds and not just areca nut. This modification is necessary as many of the beneficial effects described cannot be directly related to areca nut, which would cause incorrect interpretations and citations in the future.
Finally, the authors mentioned the carcinogenic effect of areca nut, classified as group 1 carcinogens by the International Agency for Cancer Research (IARC), for the risk of oral and oesophageal cancers in a very superficial way. The authors should add a topic describing this action and comparing the beneficial effects of the areca nut and the risk of its carcinogenic effect. The authors also need to rewrite the limitations of the study.
Round 2
Reviewer 1 Report
Comments and Suggestions for Authors
The authors have well addressed on all the issues.